# What are the effects of teaching Evidence-Based Health Care (EBHC) at different levels of health professions education? An updated overview of systematic reviews

Malgorzata M. Bala[ID]$^{1\odot}$*, Tina Poklepović Peričić$^{2\odot}$, Joanna Zajac$^1$, Anke Rohwer$^3$, Jitka Klugarova[ID]$^4$, Maritta Välimäki$^{5,6}$, Tella Lantta[ID]$^5$, Luca Pingani$^{7,8,9}$, Miloslav Klugar$^4$, Mike Clarke$^{3,10}$, Taryn Young$^3$

1 Chair of Epidemiology and Preventive Medicine, Jagiellonian University Medical College, Krakow, Poland, 2 Department of Research in Biomedicine and Health, University of Split School of Medicine, Split, Croatia, 3 Centre for Evidence-based Health Care, Division of Epidemiology and Biostatistics, Department of Global Health, Stellenbosch University, Cape Town, South Africa, 4 The Czech National Centre for Evidence-Based Healthcare and Knowledge Translation (Cochrane Czech Republic, Czech CEBHC: JBI Centre of Excellence, Masaryk University GRADE Centre), Institute of Biostatistics and Analyses, Faculty of Medicine, Masaryk University, Brno, Czech Republic, 5 University of Turku, Department of Nursing Science, Turku, Finland, 6 Xiangya Center for Evidence-based Nursing Practice & Healthcare Innovation, A JBI Affiliated Group, Central South University, Changsha, Hunan, China, 7 Department of Biomedical, Metabolic Sciences and Neurosciences, University of Modena and Reggio Emilia, Modena, Italy, 8 Department of Health Professions, Azienda USL-IRCCS di Reggio Emilia, Reggio Emilia, Italy, 9 GIMBE Foundation, Bologna, Italy, 10 Northern Ireland Clinical Trials Unit and Methodology Hub, Centre for Public Health, Queen's University Belfast, Belfast, United Kingdom

$\odot$ These authors contributed equally to this work.
* malgorzata.1.bala@uj.edu.pl

**Data Availability Statement:** All relevant data are in the paper and its S1–S3 Files and S1 and S2 Datas and S1 and S2 Checklists and S1 Table.

## Abstract

### Background

Evidence-based healthcare (EBHC) knowledge and skills are recognised as core competencies of healthcare professionals worldwide, and teaching EBHC has been widely recommended as an integral part of their training. The objective of this overview of systematic reviews (SR) was to update evidence and assess the effects of various approaches for teaching evidence-based health care (EBHC) at undergraduate (UG) and postgraduate (PG) medical education (ME) level on changes in knowledge, skills, attitudes and behaviour.

### Methods and findings

This is an update of an overview that was published in 2014. The process followed standard procedures specified for the previous version of the overview, with a modified search. Searches were conducted in Epistemonikos for SRs published from 1 January 2013 to 27 October 2020 with no language restrictions. We checked additional sources for ongoing and unpublished SRs. Eligibility criteria included: SRs which evaluated educational interventions for teaching EBHC compared to no intervention or a different strategy were eligible. Two reviewers independently selected SRs, extracted data and evaluated quality using

**Funding:** MK and JK were supported by grant number LTC20031 – "Towards an International Network for Evidence-based Research in Clinical Health Research in the Czech Republic" from Ministry of Education, Youth and Sports (INTER-EXCELLENCE) in Czech Republic and EVBRES (EVidence-Based RESearch) COST Action (CA-17117). All other authors have not received any funding for the research work reported in the article. MMB, TPP, JZ, JK, LP, MV, TL, MK worked on the review as part of their activities within Working Group 2 Activity 5 of EVBRES COST Action (CA-17117) (evbres.eu) chaired by prof. Hans Lund (Western Norway University of Applied Sciences, Bergen, Norway). The publication fee for this paper is planned to be supported from EU COST Action "EVidence-Based RESearch" number CA-17117. The sponsors had no role in the study design, data collection, analysis, interpretation of the data, writing the report and in the decision to submit the paper for publication.

**Competing interests:** All authors have completed the ICMJE uniform disclosure form at ww.icmje. org/coi_disclosure.pdf. I have read the journal's policy and the authors of this manuscript have the following competing interests: Publication fee for this paper is planned to be supported from EVBRES Cost Action (CA-17117); JK and MK reported support for this work from EVBRES Cost Action CA-17117 and Ministry of Education, Youth and Sports (INTER-EXCELLENCE) in Czech Republic (grant number LTC2003); other authors have not received any support for the research work; MMB, TPP, JZ, JK, LP, MV, TL, MK are part of EVBRES Cost Action CA-17117, which supports travel and meetings; no other relationships or activities that could appear to have influenced the submitted work.

standardised instrument (AMSTAR2). The effects of strategies to teach EBHC were synthesized using a narrative approach. Previously published version of this overview included 16 SR, while the updated search identified six additional SRs. We therefore included a total of 22 SRs (with a total of 141 primary studies) in this updated overview. The SRs evaluated different educational interventions of varying duration, frequency, and format to teach various components of EBHC at different levels of ME (UG, PG, mixed). Most SRs assessed a range of EBHC related outcomes using a variety of assessment tools. Two SRs included randomised controlled trials (RCTs) only, while 20 reviews included RCTs and various types of non-RCTs. Diversity of study designs and teaching activities as well as aggregated findings at the SR level prevented comparisons of the effects of different techniques. In general, knowledge was improved across all ME levels for interventions compared to no intervention or pre-test scores. Skills improved in UGs, but less so in PGs and were less consistent in mixed populations. There were positive changes in behaviour among UGs and PGs, but not in mixed populations, with no consistent improvement in attitudes in any of the studied groups. One SR showed improved patient outcomes (based on non-randomised studies). Main limitations included: poor quality and reporting of SRs, heterogeneity of interventions and outcome measures, and short-term follow up.

## Conclusions

Teaching EBHC consistently improved EBHC knowledge and skills at all levels of ME and behaviour in UGs and PGs, but with no consistent improvement in attitudes towards EBHC, and little evidence of the long term influence on processes of care and patient outcomes. EBHC teaching and learning should be interactive, multifaceted, integrated into clinical practice, and should include assessments.

## Study registration

The protocol for the original overview was developed and approved by Stellenbosch University Research Ethics Committee S12/10/262.

## Update of the overview

Young T, Rohwer A, Volmink J, Clarke M. What are the effects of teaching evidence-based health care (EBHC)? Overview of systematic reviews. PLoS One. 2014;9(1):e86706. doi: 10.1371/journal.pone.0086706.

## Introduction

Evidence-based health care (EBHC) is accepted by many as an approach for improving health care [1]. Implementing the principles of EBHC is important in providing continuous improvements in the quality and safety of delivered healthcare. It requires lifelong self-directed learning about the fundamentals of research and application of an evidence-based approach to resolve clinical and other healthcare problems [2]. The Lancet report on the 21st century health profession highlights the importance of EBHC knowledge, skills and attitudes and suggests a shift to transformative learning in the training of healthcare professionals, where

memorization of facts would be replaced with "critical reasoning that can guide the capacity to search, analyze, assess and synthesize information for decision-making" [3].

Teaching and learning of EBHC has been widely recommended as an integral part of the training of healthcare professionals [4, 5] with EBHC knowledge and skills recognized as core competencies of healthcare professionals worldwide [3, 6]. Teaching EBHC covers the process of practicing EBHC that includes: i) identifying knowledge gaps and formulating focused questions; ii) designing search strategies and identifying appropriate evidence to answer the questions; iii) critically appraising and interpreting research findings; iv) understanding the applicability and generalizability of research findings; and v) monitoring and evaluating performance [7]. Many EBHC teaching and learning strategies have been implemented, including face-to-face, online and blended learning, that may involve directed or self-directed learning, and may be delivered using different modalities, such as journal clubs, bed-side teaching, lectures or workshops [6, 8]. Other strategies include interactive teaching with problem-based learning, sharing information, flipped classrooms, group work, seminars coupled with discussions, oral student presentations, different forms of experiential learning, as well as interdisciplinary collaboration with librarians and through different kinds of interactive and clinically integrated teaching strategies. Newer methods, focused on gaming and simulation techniques (virtual and no-virtual), have also been implemented and, in clinical practice, mobile devices have been used for finding information, critical appraisal of clinical guidelines or specific task-oriented information in relation to clinical practice [9]. However, EBHC is still reported as being not sufficiently integrated into curricula for health professionals, for example in nursing [10].

According to the three level hierarchy of EBHC teaching and learning strategies developed by Khan et al., interactive teaching that includes clinical work is considered the most effective way to teach EBHC, ahead of classroom didactics or stand-alone teaching [11]. In line with this hierarchy, there is evidence suggesting that a standalone EBHC course, not integrated into the larger clinical curriculum, is unlikely to be successful in achieving the expected knowledge and behavioural changes, and that skills, attitudes, and behaviours improve more if the learning of EBHC is integrated in a clinical context, compared to using traditional didactic methods [12]. Combining small group discussions, e-learning, case-based teaching or computer lab sessions with didactic lectures is considered useful for achieving the intended knowledge and skills outcomes [8, 13, 14]. However, integrating all five steps when teaching EBHC, including decision making based on the best available research evidence, clinical expertise and patient preference may be challenging [15, 16]. It is influenced by the timing of delivery of EBHC education, [13, 17] difficulties with motivating students to learn EBHC, [13, 18] applying the most suitable level of clinical integration [17] and the most appropriate theoretical framework for achieving changes in knowledge, skills and attitudes, as well as the development of EBHC related changes in behaviour [19, 20].

An overview of systematic reviews assessing the effects of teaching EBHC, published in 2014, concluded that EBHC teaching and learning strategies should be multifaceted, integrated into the clinical context and should include assessment [21]. Since then, several systematic reviews addressing questions related to teaching EBHC have been published. However, evidence on the optimal learning environment, background and learning style of the learners, delivery format, and structure of the most optimal course is still lacking [22]. An update of the overview was needed to follow up on the SRs conducted after the publication of the original overview, which provided clear guidance for future studies about the target interventions, populations, and outcomes, along with preferred ways of measuring them and the required follow-up time, was published. Assessing the subsequent evidence and adding up the available findings will provide a clearer understanding of what works for whom and under which

circumstances. We decided to follow the approach of the 2014 overview and to use the already existing evidence from available SRs instead of duplicating the effort and generating needless research waste. We therefore aimed to update the overview published in 2014 to assess the most recent evidence on the effects of various approaches used in teaching EBHC to healthcare professionals at undergraduate and postgraduate level on changes in knowledge, skills, attitudes and behaviour.

## Methods

This is an update of the overview published in 2014, [21] for which the protocol was developed and approved by Stellenbosch University Research Ethics Committee S12/10/262 (S1 File). We followed the methods specified for the previous version of the overview but modified the search for this update. We followed PRISMA [23] and the Synthesis Without Meta-analysis (SWiM) [24] guidelines to report the methods and findings of this overview (S1 and S2 Checklists).

### Criteria for considering systematic reviews for inclusion

In line with the original overview, [21] this update included systematic reviews (SRs) of randomized trials, quasi-randomized trials, controlled before-and-after studies and interrupted time series studies in which any pedagogical approach intended to teach any component of EBHC methods and principles (such as the specific process of asking questions, acquiring and assessing the evidence, and considering their applicability) was compared to no intervention or to a different pedagogical approach. Systematic reviews evaluating educational activities delivered through undergraduate and postgraduate courses, and to healthcare professionals, were eligible. Outcomes of interest for this overview included knowledge, skills, attitudes and practice related to EBHC. Systematic reviews published from 2013 to 2020 were added for this update, regardless of language or publication status. Eligible SRs had to have predetermined objectives and predetermined criteria for eligibility (a protocol), have searched at least two data sources (including at least one electronic database), and have performed data extraction and risk of bias assessment of included studies. When no information about the protocol was provided in the article, we checked in PROSPERO and contacted the authors via e-mails.

### Search methods for identification of systematic reviews

We searched Epistemonikos (Epistemonikos. Epistemonikos Foundation, Arrayán 2735, Providencia, Santiago, Chile; available at https://www.epistemonikos.org) to identify eligible SRs. Epistemonikos is based on searches of a number of relevant databases for systematic reviews (https://www.epistemonikos.org/en/about_us/methods), including: Cochrane Database of Systematic Reviews, PubMed, EMBASE, CINAHL, PsycInfo, LILACS, Database of Abstracts of Reviews of Effects, The Campbell Collaboration online library, the Joanna-Briggs Institute (JBI) database of Systematic reviews and Implementation Reports (JBI Evidence Synthesis), and the EPPI-Centre Evidence Library. We limited our searches for this update to Epistemonikos rather than using the individual databases searched in the previous overview because these are now covered in Epistemonikos, single reliable database. This approach allows reducing the time and resources, as there is no need of duplicates removal, and no influence of possible differences on search terms in individual databases.The MEDLINE search strategy used in the original overview was adapted for the search of Epistemonikos (S2 File). Terms used in the MEDLINE search as index terms and text words were used for searches in Epistemonikos as text words (for example, 1. evidence based healthcare, evidence based medicine, evidence based practice, also with specific medical fields; 2. medical education, teaching, learning,

instructions, education) in the titles and abstracts of the records. Instead of keywords for systematic reviews, we used the filters for systematic review available in Epistemonikos. There were no restrictions on language of publication. Publication type and Date of publication filters were applied to restrict the search to SRs published from 1 January 2013 to the date of the search (27 October 2020). Ongoing SRs were identified from searches of PROSPERO, Cochrane Database of Systematic Reviews, JBI Evidence Synthesis, Campbell Library and The Best Evidence Medical Education (BEME) Collaboration. Backwards searching was conducted to check for potentially eligible reviews not identified through database searching.

## Study selection and data collection

Results of the search were exported from Epistemonikos into Covidence (Covidence systematic review software, Veritas Health Innovation, Melbourne, Australia. Available at www. covidence.org) and duplicates were removed. Four authors (MMB, JK, TPP, JZ) screened titles and abstracts independently and in pairs. Full texts of potentially eligible articles were obtained and, following calibration exercises, full texts were screened for inclusion by the authors (TPP, MMB, JK, JZ, LP, MV, TL, MK) and collaborators listed in the Acknowledgement section (MP, GT), working independently in pairs. Disagreements arising at any stage of the study selection process were resolved by an arbiter reviewer (TPP, MMB, AR, TY) not involved in the original assessments. Before the start of data extraction, we piloted the data extraction form used in the original overview [21] on one review to ensure common understanding of the extracted items among all authors. Seven reviewers (including authors: TL, MV, LP and collaborators: MPB, MP, GT, DL) worked in pairs to extract data from the included SRs (as specified in the original overview) into a form developed in Excel (version Office 2012, Microsoft Corporation, Redmond, WA, SAD). The extracted data included specific details about the populations involved, review methods, interventions (or concept), controls, outcomes and key findings relevant to the objectives of this overview. All extractions were checked by three authors (TPP, JZ, JK). Any disagreements were resolved by discussion. Authors of the papers were contacted if missing or additional data were required. We also searched PROSPERO to identify protocols of the reviews not providing any information about protocol, and have contacted corresponding authors regarding the availability of the protocol via emails.

## Assessment of methodological quality

The systematic reviews newly identified in this update were critically appraised using the standardized critical appraisal instrument AMSTAR 2 (A MeaSurement Tool to Assess Reviews) [25]. In the original version of the overview, methodological quality of included SRs was assessed using AMSTAR [26] which contained 11 items. Overall, according to the AMSTAR, quality of the SRs in the original version of the overview was classified as high if they met between 8 and 11 criteria, medium quality–if they met 4 to 7 criteria and low–if they met 3 or less criteria.

In this update we revised critical appraisal of studies included in the previous version of the overview by using the updated tool. AMSTAR 2 is a revised AMSTAR tool that aims to assess the quality of systematic review methods in relation to 16 distinct criteria, with each item of the AMSTAR 2 judged with "Yes", "No"and "Partial Yes"[25]. Seven of these 16 are specified as critical domains: items 2 (protocol), 4 (literature search), 7 (justification for excluding studies), 9 (risk of bias of individual studies), 11 (synthesis methods), 13 (incorporation of risk of bias in interpretation), and 15 (publication bias). Six reviewers familiar with AMSTAR 2 (MMB, JK, TPP, AR, JZ, MK) assessed methodological quality by working independently in pairs. Upon completion, assessments were analysed against each other and disagreements

were resolved by discussion. Overall quality of included SRs was judged by adhering to the published guidance with the following criteria: high–for having none or only one non-critical weakness, moderate—if there was more than one non-critical weakness, low–one critical flaw, critically low–more than one critical flaw [25].

## Data synthesis

We initially planned to report the effects of strategies to teach EBHC using relevant effect measures along with 95% confidence intervals (CI). However, due to the high heterogeneity of interventions, comparators, participant groups and methods used to measure the effects in the included SRs, as well as different outcomes being addressed, quantitative synthesis was not possible. Furthermore, most SRs had missing or poorly reported effect sizes of included studies, which made it impossible for us to undertake a meta-analysis of the effect estimates. Therefore, we reported the review level findings as provided in the included SRs and presented the results narratively for all SR included in the overview. Our descriptive summary followed the approach used in the original overview and considered grouping the findings according to the (i) population (undergraduate, postgraduate, and mixed), (ii) educational interventions versus no intervention, control intervention, or pre-test, and (iii) outcomes including knowledge, attitudes, skills, composite outcomes, behaviour changes, use in clinical practice and patient outcomes. Effect measures were used when available. This updated overview followed the conceptual framework for identifying "what works, for whom, under which circumstances and to what end" (Table 1). Because of high variability in questions asked by each included review, populations, interventions, and comparators analysed, as well as the way the outcomes were measured, we were unable to further explore the heterogeneity of the effect sizes as they were not sufficiently reported by the SRs. Insufficient information provided in the included SRs prevented us from assessing the certainty of the reviews' findings. Therefore, we followed the guidance from the Cochrane Handbook [27] and presented the certainty of evidence assessments as they were reported in the included SRs. However, only two included reviews [28, 29] reported the certainty of evidence.

To collect more detailed information on interventions and their effects, besides having included data from the SRs, we also checked the full texts of all studies included in the SRs. The individual study level findings are described here per study design and the level of medical

**Table 1. Conceptual framework for data synthesis [21].**

| |
|---|
| What works? |
| Objectives |
| Interventions |
| Methods of teaching |
| For Whom? |
| Population targeted by the intervention |
| Under which Circumstances? |
| Setting |
| Duration |
| Frequency of the intervention |
| To what end (i.e., desired outcomes) |
| Knowledge and awareness (short-term) |
| Attitude (medium-term) |
| Practice (long-term) |

education. We made no attempts to quantitatively summarise the findings or assess the quality at the level of the included studies.

We used tabular and graphical methods to present the findings at the review level (as reported by the authors of the review) and on the individual study level findings, including information on different population groups, interventions used, and relevant outcomes assessed.

We used colour coding corresponding to the different directions of effect: dark green (consistent improvement reported by all reviews in a comparison, or all studies if only a single review was included for the comparison), light green (less consistent improvement (improvement found in some reviews/ studies/ certain designs but not in all reviews/ studies/ designs, or improvement found only in a single study with weak design (BA)), yellow (reviews included in the comparison or studies included in the review in the case of a single review reported no difference between the groups), grey (unclear, inconsistent results) and white (not assessed). Individual reviews, not the primary studies included in reviews, were used as the unit of analysis unless only a single review was available for a specific comparison. In coding the colours, we did not take account of any overlap of the primary studies. Explanations of the colour coding are provided alongside tables and figures to allow better understanding of the summarised data.

## Results of the updated search

The updated search yielded 1110 references. After duplicates were removed, 1062 titles and abstracts were screened, of which 1005 were deemed irrelevant. We obtained and screened full texts for the remaining 57 records, of which 50 were excluded, leaving seven new SRs eligible for inclusion. One of these newly identified SRs [30] was the final publication of a SR that was included as an unpublished version in the original overview [31]. Therefore, together with the SRs identified in the original overview (n = 16), a total of 22 SRs are included in this update. Lists of the excluded reviews with reasons for exclusion is provided in S1 Table. We received no response from the majority of authors whom we contacted for more information regarding the protocol. Of those who responded, they either did not develop a protocol in the first place, or they worked according to a methods plan which they destroyed after they had finished their research. One author reported having worked according to the protocol of another SR. Fig 1 illustrates the process of selecting SRs for inclusion in the update, including the previously included SRs and the results of the updated search.

## Description of included systematic reviews

This overview includes a total of 22 SRs: 16 identified in the original overview [12, 28, 30–44] and six new SRs [8, 29, 45–48] (Table 2A and 2B). One SR [33] was published in French and all others were published in English. From the two SRs identified as ongoing in the 2014 overview, one has since been published and included in this update,[8] but no information is available on the second [49]. Two further reviews are awaiting assessment [50, 51]. At the time of this update, we also identified one ongoing SR, registered at BEME Collaboration [52]. Considering all 22 SRs included in this update, two included only randomized controlled trials (RCT), while 20 included RCTs and various types of non-randomized study, such as non-randomized controlled trials (CT), controlled before-after studies (CBA) and before-after studies (BA). We have focused on the findings from RCTs, CTs, CBAs and BA studies.

Dates of publication ranged over 27 years. The first SR was published in 1993, followed by six more up to 2006, and then one or two per year. The most recent SR was published in 2019. Two focused on undergraduate students, [30, 31, 47] 11 on both under- and postgraduates [8,

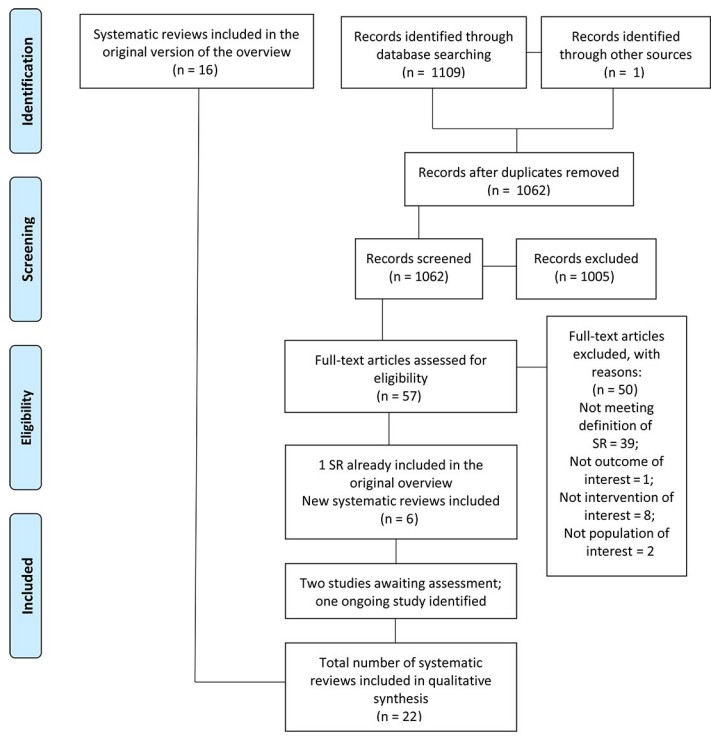

From: Moher D, Liberati A, Tetzlaff J, Altman DG, The PRISMA Group (2009). *Preferred Reporting Items for Systematic Reviews and Meta-Analyses: The PRISMA Statement*. PLoS Med 6(7): e1000097. doi:10.1371/journal.pmed1000097

For more information, visit www.prisma-statement.org.

**Fig 1. Flow diagram on the selection of systematic reviews.**

28, 33, 34, 38–42, 44, 45] and nine on postgraduates and practicing healthcare professionals [12, 29, 32, 35–37, 43, 46, 48]. The SRs evaluated a variety of educational interventions of different formats (lectures, tutorials, instructions, problem-based learning, small group discussions, journal clubs, workshops, clinically integrated methods delivered online or face-to-face or in a blended form), duration and frequency and covered various components of EBHC Table 2A and 2B). We categorized interventions into single interventions, covering a single educational activity, such as a journal club, workshop, lecture or e-learning; and multifaceted interventions, which include a combination of educational strategies such as journal clubs, tutorials, mentoring, lectures, seminars, e-learning, for example a combination of tutorials and journal clubs, tutorials and practical mentoring, lecture and small group session and computer practice. Most SRs assessed a range of outcomes, which could be categorised as knowledge, skills, attitudes, and behaviour, and in many cases were related to critical appraisal. Two SRs focused on patient outcomes and process of care [29, 48]. For two other SRs, these outcomes were also pre-specified, but not reported in the included primary studies. Across the 22 SRs, a wide variety of outcome assessment tools were used.

## Methodological quality of systematic reviews

The quality of included SRs varied significantly (Fig 2). Based on AMSTAR 2 three were judged as having high quality [8, 28, 43], one as moderate quality [39], one as having low quality [29] and seventeen as having critically low quality [12, 27, 30, 32–38, 41, 42, 44–48]. The most common reasons for low quality were not using satisfactory risk of bias assessment in

**Table 2.** A. Characteristics of included systematic reviews: Undergraduate (UG) and postgraduate (PG), reviews shaded in grey were newly identified. B. Characteristics of included systematic reviews: Postgraduate (PG) and continuing professional development in healthcare professionals (HCPs). Studies shaded in grey were newly identified.

**A**

| Review ID | Types of participants | Interventions | Comparisons | Studies included | Outcomes | Date recent search |
|---|---|---|---|---|---|---|
| Audet 1993 [33] | UG (Medical students); PG (Residents) | Lectures (weekly), Journal clubs, Once-off sessions, Biostatistics module | Not specified | 2 RCTs (post-test only); 3 non-RCTs; 3 CBAs; 1 BA; 1 CS | Knowledge; Reading habits; Critical appraisal skills | Included studies 1980–1990 |
| Ahmadi 2015 [30] (in original overview Baradaran 2013 [31]) | UG (medical, clinical and osteopathic medicine students, clinical clerks, interns at foundation year 1 and 2) | Clinically integrated methods; Short instructions; E-learning; PBL; Other multifaceted interventions | No intervention, EBHC course alone, Computer-assisted modules (self-directed learning), lecture, usual (not PBL/face-to-face) teaching | 10 RCTs; 6 non-RCTs; 11 BAs | Knowledge (asking, appraising, EBHC); Skills (asking, appraising, EBHC) attitudes, behaviour | May 2011 |
| Deenadayalan 2008 [34] | UG, graduates, PG and HC professionals (clinicians: obstetrics and gynaecology; clinical epidemiology and biostatistics; internal medicine; general surgery; emergency medicine; mental health; psychiatry; nursing; geriatric medicine) | Journal clubs | Any | 3 RCTs; 3 CTs; 2 cohorts; 3 curriculum reports; 5 reports; 1 interventional study; 1 review of journal club; 1 feasibility study; 1 personal report; 1 pilot study | Knowledge (current medical literature; Research methods; Statistics); Reading habits; Critical appraisal skills | Not reported |
| Harris 2011 [38] | UG (multidisciplinary); PG (community medicine; internal medicine; ophthalmology; surgery; emergency medicine; obstetrics and gynaecology; psychiatry) | Journal clubs in different formats | Not clearly described | 1 RCT; 1 CT; 2 non-RCTs; 8 BAs; 6 surveys | Knowledge; Reading behaviour; Confidence in critical appraisal; Critical appraisal skills; Ability to apply findings to clinical practice | Not reported |
| Hecht 2016 [45] | UG (nursing, occupational therapy, physiotherapy students); HC professionals (nurses; occupational therapists; physiotherapists; speech pathologists; dieticians; social workers; physicians; librarians) | EBHC training programs: classroom-based activities or co-intervention in addition to classroom teaching (mentorship, online support, email lists to facilitate communication / presentation of relevant literature in clinical settings) | Control group | 4 RCTs; 2 non-RCTs; 7 BAs | EBHC knowledge, attitudes; skills; Increased EBHC uptake/implementation/implementation behaviour | September 2014 |
| Horsley 2011 [28] | UG (Interns in internal medicine), PG (HC professionals) (general practitioners, hospital physicians, professions allied to medicine, and healthcare managers and administrators, surgeons) | Journal club and a workshop (0.5 day); critical appraisal materials, group discussions, articles; workshop (0.5 day) based on a Critical Appraisal Skills Programme | Standard conference series on ambulatory medicine; Access to journals and articles only; waiting list for workshop | 3 RCTs | Knowledge; Critical appraisal skills | January 2010, June 2011 |
| Hyde 2000 [39] | UG (Medical students; Interns); PG (Residents; Midwives; physicians, managers and researchers) Multidisciplinary (qualified doctors, managers and researchers) | Critical appraisals skills (Lecture, Tutorial, Seminar, Workshop, Study day, Journal club) | Not specified | 1 RCT; 8 CTs; 7 BAs | Knowledge; Skills; Behaviour; Attitude | December 1997 |

(*Continued*)

**Table 2.** (Continued)

| | | | | | | |
|---|---|---|---|---|---|---|
| Ilic 2009 [40] | UG (medical, nursing, allied health professionals) PG (general practitioners, medical residents, general surgeons, allied health professionals) | Tutorials; Workshop (0.5 day; 7 week-2hour); Multimedia package; Supplemented EBHC teaching (directed vs. self-directed) | Not described for all studies; Alternative clinical topics; Directed vs. self-directed learning | 3 RCTs; 1 CT; 1 non-RCT; 1 cross-over trial; 1 BA | EBHC knowledge; skills behaviour; Skills (Critical appraisal, formulating questions; Searching); EBP competency | September 2008 |
| Norman 1998 [41] | UG (medical residents); PG (residents) | UG: EBHC teaching in internal medicine clerkship (part of course credit); PG: journal club in variety of format | Not specified | 3 RCTs; 6 CTs (1 with cross-over); 1 cohort with historical controls | Knowledge, skills; use of the literature (Self-reported) | Searched for studies between 1966 and 1995 |
| Ramis 2019 [47] | UG (Medicine, pharmacy, nursing, and nutrition) | Theory-based EBHC teaching strategies: didactic lectures, small group discussions, facilitated workshops and PBL activities | Control, not further specified | 2 quasi-experimental; 2 BA; 1 mixed-methods design with CBA and qualitative | EBHC knowledge; skills; attitudes; behaviour; self-efficacy (or self-confidence); beliefs; values; EBHC use or future use | December 2016 |
| Rohwer 2017 [8] | UG (medical, nursing) PG (physicians, residents; nurses, practicing nurses; Physiotherapists; Physician assistants; Athletic trainers; Non-specified combination of HC professionals, clinicians, methodologists, policy makers and trainees) | Pure e-learning; Blended learning | No EBHC learning; Face-to-face learning; pure e-learning; pure e-learning with different components | 13 RCTs, 7 cluster RCTs, 4 quasi-randomized trials | EBHC knowledge; knowledge and skills as composite outcome; skills; attitude; behaviour; Process outcomes: Satisfaction with learning; Enablers and barriers of EBHC learning; Attrition of learners | May 2016 |
| Taylor 2000 [44] | UG (Medical students), PG (newly qualified physicians) | Trainings in critical appraisal skills—various educational interventions of various durations (from 180 min/one-week period to 16h/one-year period) | No educational input; general medical input; traditional epidemiological education | 1 RCT, 8 non-RCTs; 1 Cross-sectional study | Knowledge (epidemiology/statistics); Attitudes towards medical literature; Ability to critically appraise; Reading behaviour | December 1997 |
| Wong 2013 [42] | UG (medical, nursing, physiotherapy, occupational therapy students); PG physiotherapy | Mixture of EBHC training based on lectures and clinically integrated, which covered different steps of EBHC (from 4 days to 1.5 years of duration) | Irrespective of the presence or absence of control groups. | 2 CTs; 5 BAs; 1 longitudinal study with four measurements. | Knowledge; Attitudes; Skills | December 2011 |
| **B** | | | | | | |
| Ahmadi 2012 [32] | Residents (surgery) | EBHC teaching | Only reported for RCTs | 1 RCT; 3 BAs; 3 surveys | EBHC knowledge, attitude, behaviour, participants' satisfaction | July 2010 |
| | | Journal club | Only reported for RCTs | 1 RCT; 3 BAs; 3 surveys and 1 observational study | Knowledge (Critical appraisal, EBHC, statistics, study design), skills (self-assessed), research productivity, participants' satisfaction | |
| Coomarasamy 2004 [12] | PG and HC professionals (CME activities) | EBHC or critical appraisal training standalone or integrated | Control or baseline before training | 4 RCTs; 7 non-RCTs; 12 BAs | Knowledge, critical appraisal skills, attitude and behaviour | April 2004 |

(*Continued*)

**Table 2.** (Continued)

| Ebbert 2001 [35] | PG students (internal medicine, paediatrics, emergency medicine, obstetrics and gynaecology; physical medicine and rehabilitation) | Journal club (a meeting in a small group to discuss journal article (s)) | No journal club, before journal club, Standard conference on topics in ambulatory care, traditional, unstructured journal club | 1 RCT; 3 non-RCTs; 1 BA; 2 cross-sectional studies | Knowledge (clinical epidemiology, biostatistics); Reading habits and use of literature in practice; Critical appraisal skills | March 2000 |
|---|---|---|---|---|---|---|
| Fiander 2015 [29] | PG (Physicians, residents, allied health practitioners) | Interventions encouraging practitioners to use Electronic Health Information (EHI) including educational interventions (multifaceted group education, interactive workshops, educational materials) and /or organisational interventions (provision of health information/ access to EHI in electronic form, via mobile device, enhance interface). Only educational interventions were included in this overview. | "usual" educational sessions with a medical librarian, communication skills workshop, organisational (printed versions of health information, desktop device, usual EHI) | With educational interventions: 1RCT, 2 cluster RCTS (total: 2 RCTs, 4 cluster RCTs) | Frequency of database use; Information-seeking consultations; changes in recommended medical practices; Compliance with clinical practice guidelines | November 2013 |
| Flores Mateo 2007 [36] | PG HC professionals; medical interns; physicians; public health physicians, surgeons, occupational therapists; fellows in critical care; general practitioners, residents; medical research, managerial and nursing staff; EBHC experts, third year medical students; | Educational presentation; Journal club; Seminars; Workshops; Course and clinical preceptor; Literature search course; Multifaceted intervention; Internet-based intervention | Not specified | 11 RCTs; 5 non-RCTs; 8 BAs | EBHC knowledge; skills; behaviour; attitudes; Use in clinical practice (Therapy supported by evidence) | December 2006 |
| Green 1999 [37] | PG (residents in internal medicine, family medicine, obstetrics and gynaecology, paediatrics, surgery, emergency medicine and inter-programme curriculum) | Critical appraisal skills training (seminars, multifaceted intervention including seminars and journal clubs, clinically integrated EBHC teaching); Comprehensive, program-wide curricular change | Not relevant for most studies, pretest-posttest design for most effectiveness studies | 18 reports of EBHC curricula (study design not specified) and 7 of these looked at the effectiveness of the curriculum: 1 RCT; 4 non-RCTs; 2 BAs | Knowledge (clinical epidemiology, critical appraisal); EBHC behaviour (self-reported) | 1998 |
| Hines 2016 [46] | Nurses | Online learning methods (self-directed; online live lectures with feedback system); Face-to-face learning, group-based active learning, taught interactive lecture including group work, blended learning | Traditional learning, self-study material, online lectures | 2 non-RCTs; 7 BAs; 1 post-test only two-group comparison | Knowledge (Research, EBHC); Skills (Critical appraisal); Critical appraisal confidence | September 2014 |

(*Continued*)

**Table 2.** (Continued)

| Horsley 2010 [43] | Residents; physicians, occupational health physicians, nurses, allied health professionals | Didactic input, hands-on practice; Lecture, input from librarian; Live demonstrations, hands-on practice; Questionnaire and written instructions with examples | No intervention, continued current usual practices or a less intensive intervention. | 3 RCTs, 1 CCT | Skills (Quality, types of questions; Success in answering questions; Behaviour (Knowledge-seeking practices); Self-efficacy | August 2008 |
|---|---|---|---|---|---|---|
| Wu 2018 [48] | Nurses; respiratory therapists; social workers; occupational therapist; dieticians; physiotherapists | Educational interventions using the EBHC process and principles: (framing PICO question, searching, analysing, appraising and implementing evidence): some used a didactic approach, others used workshops, mentors or a project in practice or a combination of approaches; duration from five 2-hr educational sessions up to 18–24 months internship | None | 12 quantitative (5 described as CBAs, 7 BAs), 3 mixed-methods studies (BAs), 3 qualitative studies | Changes in patient outcomes, project-related, such as changes in pain management; the rates of urinary catheter utilisation, pressure ulcers, infection of dialysis catheters, central-line related infections, aspiration pneumonia, ventilator-associated pneumonia; the length of stay in hospital; the number of calls to the outpatient clinic; cost; anxiety of patients | May 2017 |

RCT: Randomized Controlled Trial; PG: Postgraduate; BA: Before After study; CT: Controlled Trial; UG: Undergraduate; CBA: Controlled Before After study; CS: cross-sectional; HC: Health Care.

individual studies included in the review, lack of a comprehensive search, not providing a list of excluded studies with justification for exclusion. In the SRs included in the original version of the review, lack of protocol was also a common reason for low quality. Among all the included SRs, only one SR [8] conducted a quantitative synthesis and investigated publication bias, while a second SR [36] performed funnel plot analysis despite having conducted no meta-analysis. The justification for not combining studies in meta-analysis in 21 reviews was high heterogeneity in the populations included, teaching methods and their comparators and differences in the methodological approach of outcomes assessment. Regarding non-critical items of AMSTAR2 those commonly not met included reporting sources of funding for primary studies included in the SRs, justification for study design selected to be eligible in the SR, description of included studies in detail. On the other hand, the majority of included SRs reported research questions including components of PICO, declaring conflict of interest, providing satisfactory explanation of heterogeneity (narrative).

## Overlap between included systematic reviews

To examine whether the six new SRs have added to the evidence base or merely duplicated previous work, the matrix mapping included studies to SRs was updated to include both the new SRs and the studies included within them (S1 Data). Collectively, the number of studies included in all 22 SRs is 141 (Fig 3), of which 83 were reported in the 16 SRs from the original overview and 58 studies were newly identified in the six SRs retrieved by the updated search. There was a considerable overlap among individual studies included in the SRs, with 60 studies included in more than one SR.

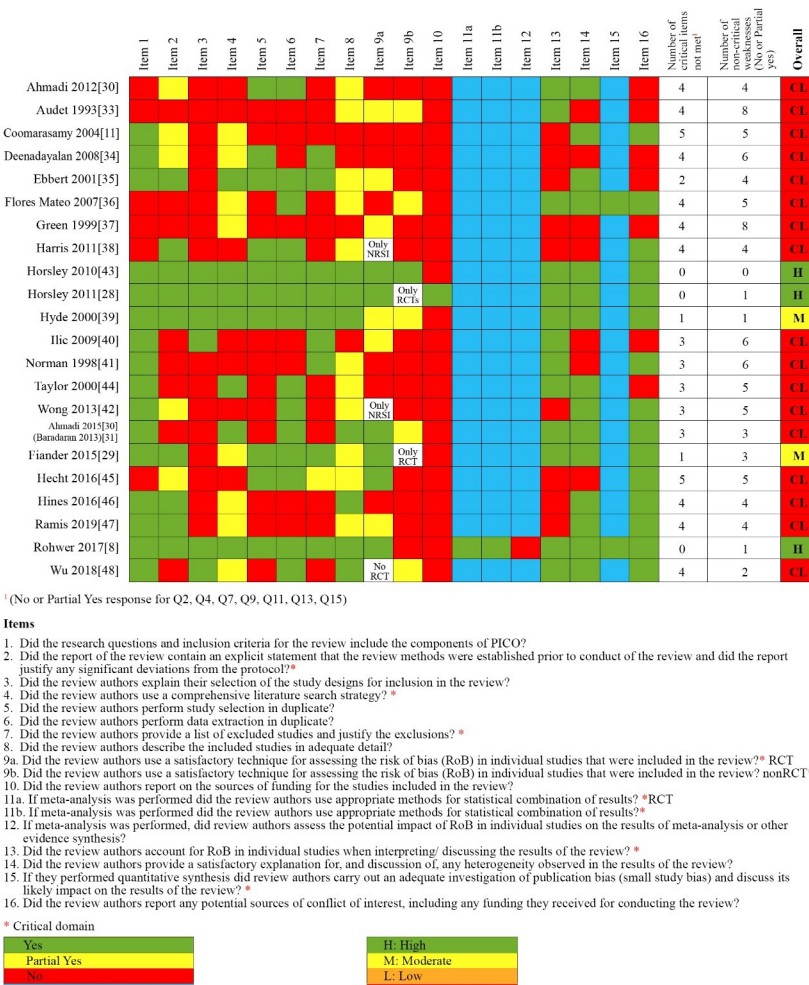

¹ (No or Partial Yes response for Q2, Q4, Q7, Q9, Q11, Q13, Q15)

**Items**

1. Did the research questions and inclusion criteria for the review include the components of PICO?
2. Did the report of the review contain an explicit statement that the review methods were established prior to conduct of the review and did the report justify any significant deviations from the protocol?*
3. Did the review authors explain their selection of the study designs for inclusion in the review?
4. Did the review authors use a comprehensive literature search strategy? *
5. Did the review authors perform study selection in duplicate?
6. Did the review authors perform data extraction in duplicate?
7. Did the review authors provide a list of excluded studies and justify the exclusions? *
8. Did the review authors describe the included studies in adequate detail?
9a. Did the review authors use a satisfactory technique for assessing the risk of bias (RoB) in individual studies that were included in the review?* RCT
9b. Did the review authors use a satisfactory technique for assessing the risk of bias (RoB) in individual studies that were included in the review? nonRCT*
10. Did the review authors report on the sources of funding for the studies included in the review?
11a. If meta-analysis was performed did the review authors use appropriate methods for statistical combination of results? *RCT
11b. If meta-analysis was performed did the review authors use appropriate methods for statistical combination of results?*
12. If meta-analysis was performed, did review authors assess the potential impact of RoB in individual studies on the results of meta-analysis or other evidence synthesis?
13. Did the review authors account for RoB in individual studies when interpreting/ discussing the results of the review? *
14. Did the review authors provide a satisfactory explanation for, and discussion of, any heterogeneity observed in the results of the review?
15. If they performed quantitative synthesis did review authors carry out an adequate investigation of publication bias (small study bias) and discuss its likely impact on the results of the review? *
16. Did the review authors report any potential sources of conflict of interest, including any funding they received for conducting the review?

* Critical domain

| Yes | | H: High |
| Partial Yes | | M: Moderate |
| No | | L: Low |
| No meta-analysis conducted | | CL: Critically low |

**Fig 2. Methodological quality of all reviews included in the update of the overview using AMSTAR 2 tool.**

## Effects of various educational interventions

The original overview [21] found that multifaceted, clinically integrated methods, with assessment, improved knowledge, skills and attitudes compared to single interventions or no interventions. Amongst residents, these multifaceted clinically integrated interventions also improved critical appraisal skills and the integration of results into patient decision making, as well as knowledge, skills, attitudes and behaviour amongst practicing healthcare professionals. Considering single interventions for residents, EBHC knowledge and attitudes were similar when comparing lecture-based teaching versus online modules. None of the SRs included in the original overview found any evidence on the effects of EBHC teaching on patient outcomes or processes of care. The update brought more evidence on the effects of different EBHC teaching methods, as compared with no intervention or control teaching methods, on knowledge, attitudes, skills, behaviour and on processes of care and patient outcomes.

**Summary of effects of interventions.** The description below contains an integrated summary of all 22 SRs included in the update (also Table 3A and 3B and S2 Data).), while the detailed description of the effects of interventions assessed in the newly included SRs in provided in S3 File.

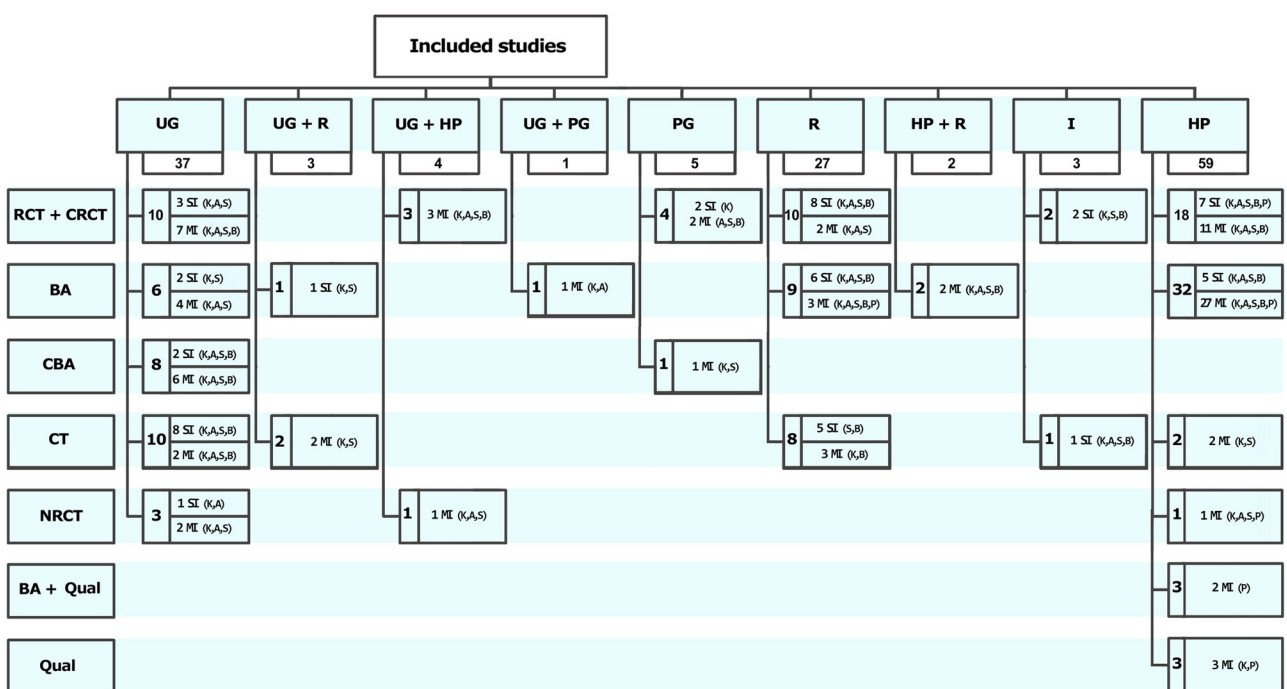

**Fig 3. Summary of source studies included in the systematic reviews.** A–attitudes, B–behaviour, BA–before and after study, CBA–controlled before and after study, CRCT- cluster RCT, CT- controlled trial, HP–health professionals, I–interns, K -knowledge, MI–multiple intervention, NRCT–nonrandomised controlled trial, P–practice, Qual–qualitative study, R–residents, RCT–randomised controlled trial, S–skills, SI–single intervention, UG–undergraduates.

Teaching critical appraisal as compared to no teaching (or pre-test) was associated with consistently reported increased knowledge across all studied populations, [28, 33, 39, 41, 44] improved skills in undergraduates [33]. Such interventions were associated with less consistently reported improved skills [28, 33, 36] and behaviour [33, 36, 41] in postgraduates, change in behaviour in undergraduates [33] as well as less consistently reported improved attitudes and inconsistent results regarding skills and behaviours in mixed populations [39, 44]. Only one study reported certainty of evidence as low for knowledge and very low for critical appraisal skills [28].

Journal clubs as compared to no journal clubs (or pre-test) were associated with consistently reported improved knowledge and behaviour and less consistently reported improvement in skills in one review covering mixed population [38]. The improvement in knowledge, attitudes, skills [32, 34–37] and behaviour in postgraduates was less consistently reported in five SRs. One SR did not report a difference when a composite score for knowledge and skills [36] and the results for EBHC use in clinical practice were not consistent in two reviews [32, 35].

E-learning as compared to no e-learning (or pre-test) was associated with consistently reported increase in knowledge, skills and attitudes in mixed populations (one SR [8]) and consistently reported increase in knowledge and skills in undergraduates (one SR [30]). Online workplace education [46] was associated with consistently improved skills in postgraduates. For blended learning compared to no learning one SR [8] reported consistent improvement in knowledge and composite score of knowledge and skills, less consistent improvement in behaviour and no effect regarding attitude in postgraduates.

Two SRs included various, mainly multifaceted interventions to teach EBHC in mixed populations [42, 45]. One SR included interventions that were mainly based in the classroom, [45]

**Table 3.** A. Review level findings: Intervention vs no intervention (22 reviews–some of the reviews are shown in more than one population if separate data were provided). B. Review level findings: Intervention vs other intervention (5 reviews).

A

| Intervention | Comparison | EBHC knowledge | Attitudes | Skills | Behaviour | Composite score (knowledge and skills) | Use in clinic practice | Patients' outcomes |
|---|---|---|---|---|---|---|---|---|
| Mixed group of participants (undergraduate, postgraduate, healthcare professionals, decision-makers, patients) (6 reviews) | | | | | | | | |
| Journal club | control/pre-test | [38] | | [38] | [38] | | | |
| Critical appraisal | control/pre-test | [39],[449] | [39], [44] | [39], [44] | [39], [4439] | | | |
| E-learning pure | no intervention | [8] | [8] | [8] | | [8] | | |
| Various, mainly multifaceted EBHC training with workshops, lectures, longer EBHC courses, small group discussions, journal club, practical sessions (classroom based), presentation, mentoring, online support, e-mail list | control/pre-test | [45] | [45] | [45] | [45] | | | |
| Various, mainly multifaceted EBHC interventions with mixed lectures and clinically integrated (majority at least 3 steps of EBHC) | control/pre-test | [42] | [42] | [42] | [42] | | | |
| Postgraduate students and healthcare professionals, managers, decision makers (15 reviews) | | | | | | | | |
| Journal Club | no intervention | [35], [34], [32], [36], [37] | [36] | [35], [34], [32], [36], [37] | [35], [34], [32], [37] | [36] | [35], [32] | |
| Critical appraisal /Critical appraisal course/ workshop, journal club, conference, presentations, reading, seminar | control/ no intervention/pre-test | [33], [28], [41] | | [33], [36], [28] | [33], [36], [41] | | | |
| Blended | no intervention | [8] | [8] | | [8] | [8] | | |
| Standalone EBHC | control/pre-test | [12], [37] | [12] | [12], [37] | [12] | | | |
| Clinically integrated /Various educational interventions supporting implementation of EBHC (lectures, seminars, workshops, mentors, fellowship, projects in practice or combinations) | control/pre-test | [12], [37] | [12] | [12], [37] | [12], [37] | | | [48] |
| Workplace education online | pre-test | | | [46] | | | | |
| Workplace education face-to-face | pre-test | [46] | | [46] | [46] | | | |
| EBHC course | control/pre-test | | | [36], [40] | | [32], [36] | | |
| Multifaceted intervention/interactive workshops | control/pre-test | [32], [36] | [36] | [32], [40], [43] | [36], [29] | [36] | | |
| Undergraduate students (5 reviews) | | | | | | | | |
| Critical appraisal course, lectures, seminars, tutorials | control/pre-test | [33], [41] | | [33] | [33] | | | |
| E-learning pure | no intervention/ pre-test | [30] | | [30] | | | | |
| Standalone short instructions (seminar, workshop, short course) | control/pre-test | [30] | [30] | [30] | [30] | | | |
| EBHC course | control | [40] | | [40] | [40] | | | |
| Clinically integrated | control/pre-test | [30] | [30] | [30] | | | | |
| Multifaceted interventions (lecture, seminar, reading, small group work, practical session, individual work, personal digital assistant; presentation, mentoring; also including various EBHC teaching interventions based on theory) | no intervention/ pre-test | [30], [47] | [30], [47] | [47] | [47] | | | |

B

*(Continued)*

**Table 3.** (Continued)

| | | | | | | | | |
|---|---|---|---|---|---|---|---|---|
| Mixed group of participants (undergraduate, postgraduate, healthcare professionals) (1 review) | | | | | | | | |
| E-learning pure | face-to-face | [8] | [8] | [8] | | | | |
| Blended | pure e-learning | [8] | | [8] | | | | |
| Blended | face-to-face | [8] | [8] | [8] | [8] | [8] | | |
| Postgraduate students and healthcare professionals, decision makers (3 reviews) | | | | | | | | |
| E-learning | lecture based clinically integrated | [32] | [32] | | | | | |
| E-learning | another e-learning | | | [8] | | | | |
| Journal club face-to-face | journal club online | | | [32] | | | | |
| Workplace education online | traditional face-to-face | [46] | | | | | | |
| Undergraduate students (2 reviews) | | | | | | | | |
| E-learning | traditional face-to-face | [30] | [30] | [30] | | | | |
| Problem-based learning EBHC teaching in small group | usual EBHC teaching (whole class) | [30] | [30] | | | | | |
| Self-directed EBHC | directed EBHC workshops | [40] | | [40] | [40] | | | |

Dark green: consistent improvement reported by all reviews in a comparison or all studies if only single review included for comparison.

Light green: less consistent improvement (improvement in some reviews/studies/certain designs but no all reviews/studies/designs or improvement only in single study with weak design (BA)).

Yellow: reviews included in the comparison or studies included in the review in case of a single review included for a comparison reported no difference between the groups.

Grey: not clear, inconsistent results.

White: not assessed.

RCT: Randomized Controlled Trial; PG: Postgraduate; BA: Before After study; CT: Controlled Trial; UG: Undergraduate; CBA: Controlled Before After study; HC: Health Care.

while the other mixed lecture-based strategies with clinically integrated teaching [42]. Various, mainly multifaceted interventions were associated with consistently increased knowledge and skills, with inconsistent findings regarding behaviour and attitudes in a mixed population. However multifaceted interventions compared to a control group or pre-test scores were associated with consistent improvement in behaviour in postgraduates, [29, 36] less consistent improvement in knowledge and skills, [32, 36, 40, 43] and no effect on attitudes in postgraduates [36]. In undergraduates multifaceted interventions compared to a control group or pre-test scores were associated with consistent improvement in skills [47] and less consistent improvements in knowledge, attitudes and behaviour [30, 47]. Only one study reported certainty of evidence as low for interventions that encouraged practitioners (physicians, residents, allied health professionals) to use electronic health information (EHI) to improve clinical practice and patient outcomes [29].

Standalone teaching of EBHC was associated with less consistently reported improved knowledge and skills and no significant effect on behaviour or attitude in postgraduates (two reviews [12, 37]). One SR in undergraduates reported that the most consistent effect was

observed for improvement in skills with less consistent effect for improvement for knowledge, attitude and behaviour [30]. An EBHC course was associated with consistently improved composite scores of knowledge and skills [32, 36] and less consistently improved skills alone in postgraduates [36, 40]; and with consistently improved knowledge and behaviour and less consistently improved skills in one SR in undergraduates [40].

Face-to-face workplace education was associated with consistently improved knowledge, skills and no difference as compared to pre-test scores for behaviour in postgraduates [46].

Clinically integrated EBHC teaching or educational interventions supporting implementation of EBHC was associated with consistently improved knowledge in postgraduates, [12, 37] and undergraduates [30], consistently improved attitudes and behaviour in two reviews in postgraduates [12, 37] and less consistently improved skills in postgraduates [12, 37] and in undergraduates [30], inconsistent results regarding attitudes in undergraduates [30] and improved patient outcomes reported in one [48] SR.

When different interventions were compared to each other, such as e-learning compared to traditional face-to-face learning, similar effects were reported for knowledge and attitudes [8, 30, 32] for all populations and for skills in undergraduates and mixed populations [8, 30]. Similar effects were also reported for computer assisted, self-directed learning of EBHC as compared to directed workshops [40] in undergraduates; and online workplace based as compared to traditional face-to-face EBHC teaching in postgraduates [46]. Face-to-face journal clubs were reported to be more effective in improving skills compared to an online journal clubs in postgraduates [32]. Blended learning compared to pure e-learning was reported to be associated with improved knowledge [8] in a mixed populations, while results regarding skills were inconsistent in one SR [8]. The same SR compared blended learning [8] to face-to-face learning and reported similar effects for knowledge and skills for both groups, consistently improved behaviour and less consistently improved attitudes.

Problem-based EBHC learning was associated with less improvements in knowledge and attitudes compared to usual EBHC teaching in undergraduates [30].

Findings from studies included in the reviews. To be able to answer the review questions according to the conceptual framework presented in Table 1, we described the findings from studies that had been included in the reviews and provided more details on the evidence around what works at which education level.

Figs 4–6 provide a detailed graphical presentation of a range of interventions assessed across study designs, including different participant groups and the observed effects, as reported in the individual studies. We used coloured symbols, namely green arrows and yellow circles, to provide a simple display of the direction of the effect, either improvement or no change in each of the specified outcomes. Explanation of the coloured symbols and abbreviations used are available alongside the figures. However, as we made no quality assessments at individual study level, and did not determine the certainty of the evidence on the outcome level, the following summary should only be regarded as a short presentation of the available interventions and the expected outcomes from the included RCTs.

A number of studies consistently showed beneficial effects of various teaching approaches at different levels of education. There is evidence from RCTs suggesting multifaceted interventions improved knowledge, skills and attitudes in undergraduate students, while findings from non-RCTs emphasized the use of small group discussions for changes in knowledge and attitudes, or e-courses for changes in knowledge of undergraduates. For health professionals, multifaceted interventions with hands-on activities on the development of PICO question and searching, as well as discussions and critical appraisal improved knowledge, behaviour and attitudes, or skills, respectively. There is also evidence in favour of clinically integrated EBHC

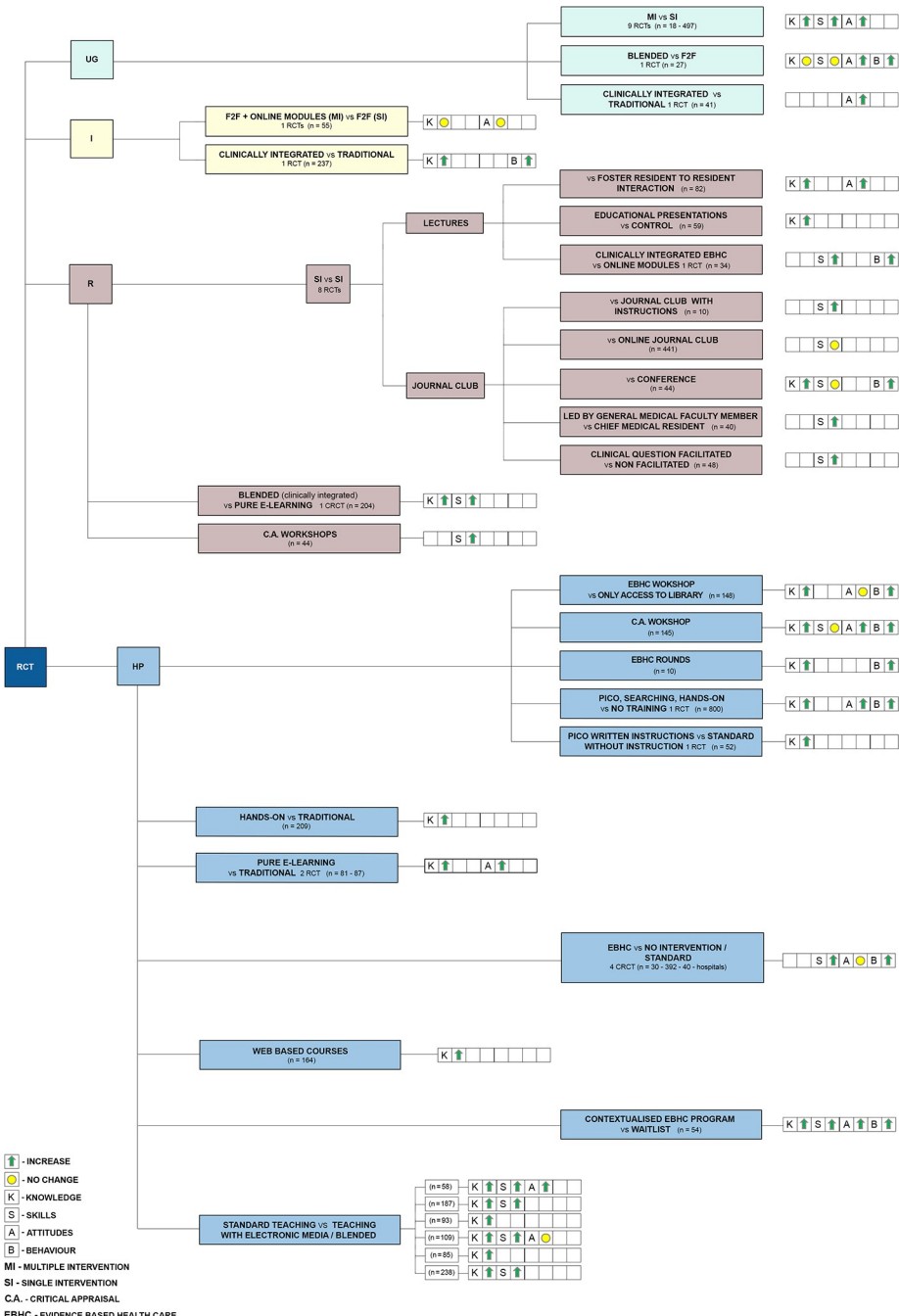

**Fig 4. Illustrative display of the likely impact of different teaching approaches at different medical education levels–summary from RCT.** HP–health professionals, I–interns, R–residents, RCT- randomised controlled trial, UG–undergraduates; Explanation of the coloured symbols used are provided on the figure.

teaching for all participant groups. Findings from RCTs reported improvements in undergraduates' attitudes, as well as in knowledge and behaviour among interns. Residents' knowledge and skills were shown to improve following a clinically integrated blended learning, while knowledge and behaviour in health professionals improved when discussions on EBHC were implemented during ward rounds.

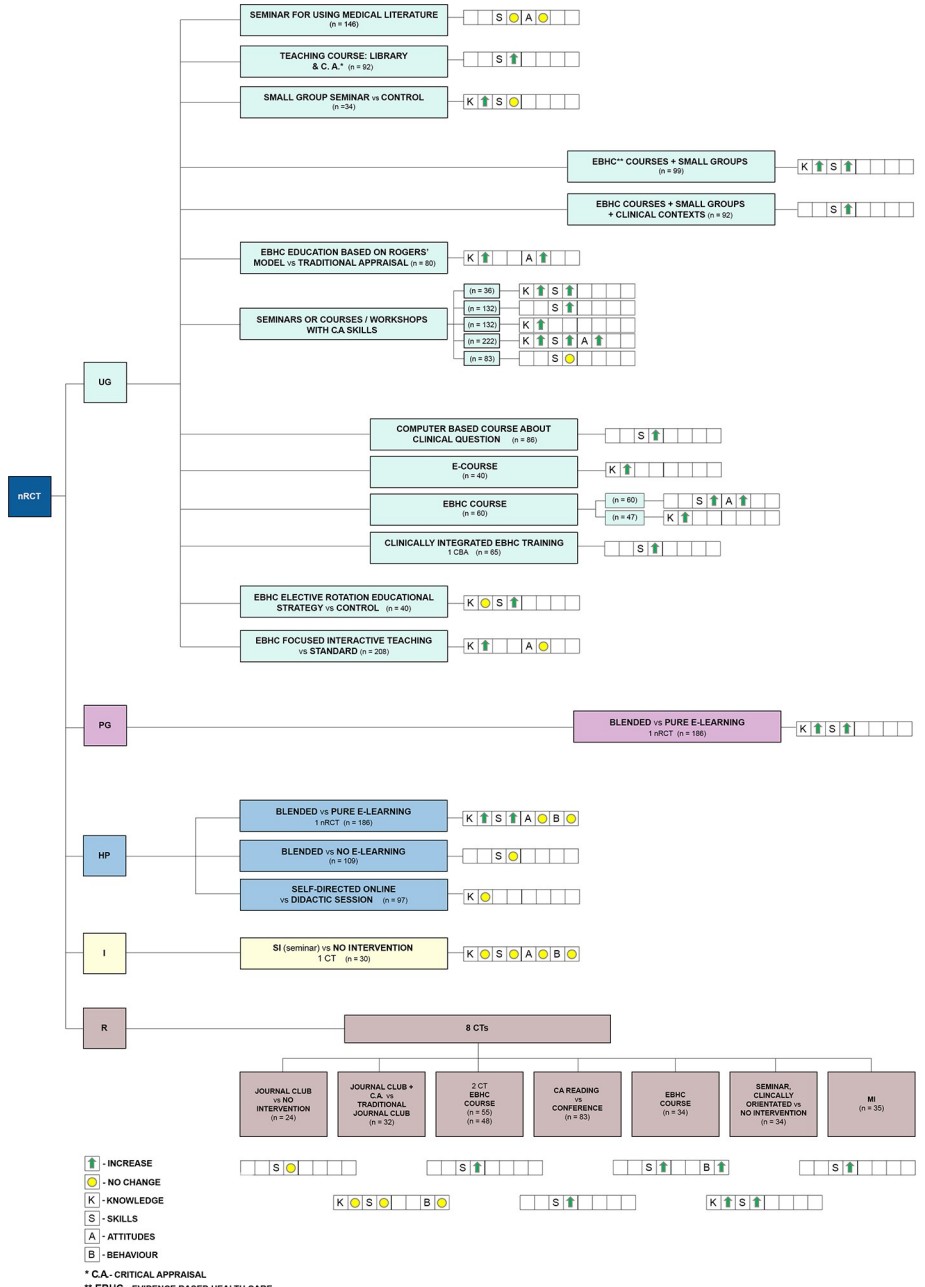

**Fig 5. Illustrative display of the likely impact of different teaching approaches at different medical education levels–summary from non-RCT.** CT–controlled trial, HP–health professionals, I–interns, MI–multiple intervention, non-RCT–non-randomised study, Qual–qualitative study, R–residents, SI–single intervention, UG–undergraduates; Explanation of the coloured symbols used are provided on the figure.

Evidence from RCTs suggested blended learning might lead to improved attitudes and behaviour among undergraduate students, or knowledge and skills among residents, either compared to traditional face-to-face learning or to pure e-learning. Likewise, blended learning was shown to be more beneficial for EBHC knowledge and skills among postgraduates as well. Number of RCTs found that blended learning in the form of standard teaching with the use of electronic online media improved health professionals' knowledge, skills and attitudes.

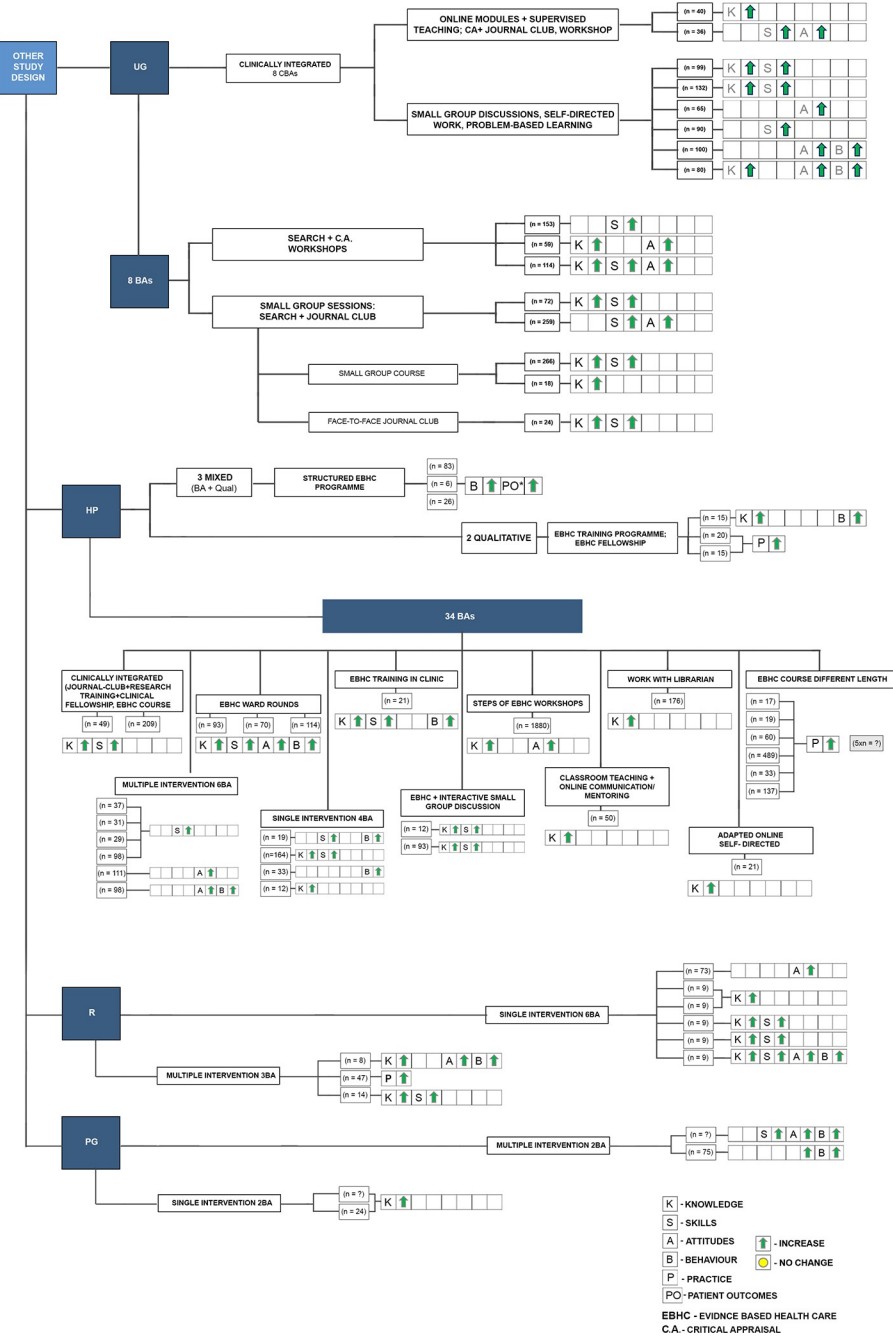

**Fig 6. Illustrative display of the likely impact of different teaching approaches at different medical education levels–summary from other study designs (i.e., before-after, qualitative studies).** BA–before and after study, CBA–controlled before and after study, HP–health professionals, R–residents, UG–undergraduates; Explanation of the coloured symbols used are provided on the figure.

Workshops on critical appraisal delivered for residents in a non-RCTs and for undergraduates in non-RCTs showed improvements in skills or skills and knowledge, respectively, along with evidence from an RCT on positive changes in knowledge, behaviour and attitudes in health professionals.

One non-RCT found that introducing interactive approach in teaching improved undergraduates' knowledge. According to the evidence from a large RCT (n = 441) there is no difference in residents' skills after a traditional or an online journal club, with evidence from an RCT suggesting pure e-learning improved health professionals' attitudes and knowledge.

A total of 34 BA studies reported that clinically integrated multiple educational interventions, especially when delivered in a clinical setting (such as during EBM ward rounds), including work in small groups and hands-on training, might improve EBHC knowledge, skills, attitudes and behaviour, and might also impact practice and contribute to the quality of care and patient outcomes. These studies also reported on changes in practice, including increased confidence in providing patient care, increased collaboration between colleagues, and a tendency to pursue a multidisciplinary approach in problem-solving among healthcare professionals following a 12-week multifaceted intervention course on EBHC.

## Discussion

### Summary of main results

This updated overview includes 22 SRs published between 1993 and 2019. These SRs included RCTs and non-RCTS, evaluated a variety of educational interventions of different formats, duration and frequency; covered various components of EBHC in a variety of settings and evaluated a range of EBHC related outcomes measured with variety of tools. The 22 SRs included a total of 141 primary studies, with 60 of these included in more than one SR, showing considerable overlap. Findings of SRs showed a consistent improvement in EBHC knowledge across all populations and interventions compared to no intervention or pre-test scores. There was also an improvement in EBHC skills, but this was less pronounced in postgraduates and less consistent in the SRs of mixed population. Systematic reviews found positive changes in EBHC behaviour in under- and postgraduates, but not in mixed populations, and no consistent improvement in attitudes in any of the studied groups. Only one SR addressed patient outcomes and this showed improvement in a variety of patient outcomes [48]. Diversity of methodological approaches and of teaching activities, as well as aggregated findings at the SR level made us unable to compare the effects of different techniques at the SR level. As expected, findings presented in the SRs were quite consistent with the findings from the individual studies, but an examination of the latter allowed more detailed comparisons and inferences about level of medical education and type of intervention. Considering only RCTs, any type of proposed educational interventions in a group of health professionals was associated with improvement in EBHC knowledge and behaviour as compared with control group receiving no intervention, while multifaceted interventions focusing on critical appraisal, methodology and discussions were effective in improving EBHC skills, compared to a control group.

For undergraduate students, all analysed interventions were associated with improved EBHC attitudes, while the effects of other outcomes were less consistent. For interns, clinically integrated educational interventions were more beneficial than traditional non-clinically integrated education interventions in increasing EBHC knowledge and behaviour. For residents, the most consistent results were achieved with blended (clinically integrated) learning compared to pure e-learning, leading to improved EBHC knowledge and skills. The data with potentially higher risk of bias from non-RCTs showed that multifaceted interventions compared to no intervention improved EBHC knowledge, skills and attitudes among undergraduate students, while blended learning showed a greater improvement in EBHC knowledge and skills in postgraduates (compared to pure e-learning) and in healthcare professionals (compared with no intervention).

The first version of this overview [21] concluded, in 2014, that 'future studies and systematic reviews should focus on minimum components for multifaceted interventions, assessment of EBHC knowledge, attitude, skills and behaviour in the medium to long term, using validated assessment tools, and on how best to implement these interventions'; and that 'further evaluation should consider the effectiveness of e-learning and the influence of various teaching and learning settings and the context within which teaching takes place.' Among the newly identified SRs, three referred to the recommendations for research of the original overview, [8, 47, 48] but only one clearly justified the conduct of the SR based on these recommendations. The studies included in the new SRs were published between 1999 and 2017, and five out of six new SRs included studies published in 2014 or later. However, the problems noted in 2014 overview regarding primary studies reporting on the intervention development, components, and implementation, using validated outcome measurements although improved, remained.

## Overall completeness, applicability and quality of evidence

Despite the wide searches in both the original overview and this update, only 22 SRs met the inclusion criteria of our overview. For this update 36 of the 46 excluded reviews, and in the total sample 39 out of 50 excluded reviews (78%) did not meet the criteria for a systematic review pre-specified in our protocol for an update and for both original overview and an update. These criteria included having predetermined objectives and predetermined criteria for eligibility (in an update specified as having protocol), having searched at least two data sources, of which one was an electronic database and having data extraction and risk of bias assessment performed. All those 39 reviews did not refer to "a priori" design or having protocol or had it registered in PROSPERO. This highlights a previously raised issue of adherence to methodological and reporting guidelines for research published as "systematic reviews" [53], and limits completeness of the evidence identified.

In many cases, the interventions included in SRs in this overview focused on a single step of EBHC, such as question formulation, searching or critical appraisal, while few focused on EBHC implementation in clinical practice. Other reviews have also highlighted that medical education often focused on teaching and assessing students on 'ask, acquire and appraise' [54, 55]. Almost all the included SRs focused on EBHC knowledge, skills, behaviour and attitudes, with only a single SR addressing processes of care or patient outcomes. These outcomes are however influenced by many factors due to the complex process [56] of translating evidence into practice. This includes: whether a healthcare professional is seeking to use best evidence, [48, 57] applicability of the evidence to the setting, availability of the interventions and related clinical pathways, and patient preferences and adherence. Therefore, having EBHC knowledge and skills, a positive attitude to EBHC, and being able to apply this (behaviour) is necessary but not sufficient to change healthcare practice. We have identified one ongoing review using a realist methodology which may provide more insights into the factors that influence the effectiveness of EBHC teaching of residents [52] in relation to outcomes and context.

In general, the SRs in this overview included a wide range of studies of various designs that had been published between 1981 and 2017. These studies assessed a variety of interventions, populations, healthcare professionals at different points of their careers and settings. However, most of the included studies were conducted in high-income countries, which may limit applicability of their findings to those teaching and learning EBHC in low- and middle-income countries (LMICs). Applicability of the findings is further limited by poor reporting of the interventions and their content [58], which hinders implementation in practice.

Kumaravel et al, who classified available EBHC tools according to the assessment of EBHC practice, the educational outcome domains that were measured and their quality and

taxonomy, found that the step 'appraise' was the most frequently assessed using a validated tool [55]. Other steps (ask, acquire, apply) were assessed less often, while the assessment of the step 'assess' was completely lacking. In our overview, some studies used validated question-naires, such as the Berlin questionnaire [59] or Fresno test [60], but many did not provide explicit information about validation of the instruments used in outcome assessment, which further limits the applicability of our findings to clinical practice. Saunders et al. also raised an issue of lack of compatibility between self-reported and objectively measured EBHC compe-tencies, with possible overestimation with self-assessments. In our overview the majority of included reviews either not specified clearly what type of instruments were used in primary studies or included a mixture of studies with both self-assessed and objectively assessed out-comes, which may further limit applicability of our findings to clinical practice [61].

Despite limiting this overview to only those SRs that met our basic criteria for a SR as pre-specified in our protocol, the quality of the included SRs was not optimal. The main shortcom-ings were not using satisfactory risk of bias assessment in individual studies included in the review an inadequate search and failure to provide a list of excluded studies with justification for exclusion and in the SRs included in the original version of the review–also lack of proto-col. Many of the SRs we included were poorly reported, with several not providing sufficient information on the characteristics of included studies and their findings.

Limits and potential biases in the overview process. To minimize risk of bias, the process of the updated overview followed standard procedures as specified in the original version of the overview. We only used more specific definition of a SR and modified the search, which was limited to Epistemonikos for this overview, rather than being performed in the individual data-bases searched in the previous overview because these are now covered in Epistemonikos. We checked additional sources for ongoing and unpublished SRs, and conducted study selection, data extraction and quality assessment in duplicate. We also followed PRISMA guidelines when reporting our methods and findings. However, our findings are limited by methodological flaws in the included SRs and, in turn, in the studies they included, heterogeneity of assessed interventions and outcome measures, and short term follow up. Since the data were presented mostly narratively in the included SRs, to further understand what works we attempted to gather more information on interventions and their effects by referring to the included studies, but we have not conducted quantitative data extraction and quality assessment of these studies due to the nature of our research, which is an overview of SRs rather than a new SR.

## Conclusions

This updated overview of SRs confirms and strengthens the findings of the previous version of the overview and shows that teaching EBHC, including e-learning, consistently improved EBHC knowledge and skills at all levels of medical education and behaviour in under- and postgraduates, while attitudes towards EBHC were not consistently improved. However, there is still little evidence on the influence of EBHC teaching on processes of care and patient outcome.

### Implications for practice

Generally, we should be teaching and learning EBHC while ensuring that it is interactive, inte-grated into clinical practice, using multifaceted interventions, and should include assessments. In addition, wider implementation of e-learning should be considered, as an adjunct to blended learning (mix of face-to-face and e-learning). Other factors such as resources, feasibil-ity and preference of learners need to be taken into consideration when planning EBHC learn-ing activities.

## Implications for research

Further SRs evaluating the effectiveness of teaching EBHC compared to not teaching EBHC are not needed. Instead, future SRs should compare various strategies to teach EBHC and should follow robust methods, including prospective registration of the titles and pre-specified protocol, comprehensive searches, proper risk of bias assessment and transparent reporting of methods and results. Future primary studies should be more robust (preferably well-designed RCTs) and should compare various strategies in various settings with a longer follow up (at least one year after completion of the course); and use validated tools for the assessment of outcomes. Studies should also aim to measure EBHC behaviour in clinical practice and patient outcomes, not just knowledge, skills and attitudes. In general, there is also a need for studies in LMIC countries as they were lacking in included SRs. The reporting of studies should also be improved, to provide sufficient information on the populations and interventions studied. Authors should consult the GREET reporting guidelines, [58] that provide guidance on the reporting of educational interventions for evidence-based health care.

## Differences between the protocol for primary overview and update of the overview

Searches were modified. The protocol for the original overview and its actual searches used a variety of electronic sources, such as the Cochrane Library (April 2013), The Campbell Library (April 2013), MEDLINE (April 2013), SCOPUS, the Educational Resource Information Center (ERIC), the Cumulative Index to Nursing and Allied Health Literature (CINAHL) (June 2013) and the Best Evidence Medical Education (BEME) Collaboration. For this update, the MEDLINE search strategy used in the original overview was adapted for a search of Epistemonikos. Additional identification of ongoing reviews was performed by searching PROSPERO, Cochrane Database of Systematic Reviews, JBI Evidence Synthesis, Campbell Library and BEME.

Eligibility criteria for the original overview were: „Systematic reviews were defined as those that had predetermined objectives, predetermined criteria for eligibility, searched at least two data sources, of which one needed to be an electronic database, and performed data extraction and risk of bias assessment." For the update the criteria were specified in more detail as follows: „Eligible SRs had to have predetermined objectives and predetermined criteria for eligibility (a protocol), have searched at least two data sources (including at least one electronic database), and have performer data extraction and risk of bias assessment of included studies. When no information about the protocol was provided in the article we checked in PROSPERO and contacted the authors via e-mails."

For the assessment of methodological quality AMSTAR2 –the most up-to-date version of AMSTAR tool was used, while in the original version of the review previous version of AMSTAR tool was used.

## Supporting information

**S1 Checklist. PRISMA checklist.**
(DOC)

**S1 Data. Matrix of included systematic reviews and the studies included in each.**
(XLSX)

**S1 File. Protocol.**
(DOC)

**S1 Table. Excluded studies with reasons for exclusion.**
(DOCX)

**S2 Checklist. SWiM checklist.**
(DOCX)

**S2 Data. Supplementary excel file with data from individual reviews.**
(XLSX)

**S2 File. MEDLINE and Epistemonikos search strategies.**
(DOCX)

**S3 File. Description of results of 6 new reviews.**
(DOCX)

## Acknowledgments

Jimmy Volmink for contribution to the original overview; Ana Utrobičić for helping with searches while preparing for the overview; Demetris Lamnisos, Marija Palibrk, Maria Brandao, Gulcan Tecirli for help with full text screening and /or data extraction, and preliminary comments on the manuscript, Anna Witkowska for graphic design of the figures in the manuscript.

## Author Contributions

**Conceptualization:** Malgorzata M. Bala, Tina Poklepović Peričić, Joanna Zajac, Anke Rohwer, Jitka Klugarova, Mike Clarke, Taryn Young.

**Data curation:** Malgorzata M. Bala, Tina Poklepović Peričić, Joanna Zajac, Anke Rohwer, Jitka Klugarova, Maritta Välimäki, Tella Lantta, Luca Pingani, Miloslav Klugar.

**Formal analysis:** Malgorzata M. Bala, Tina Poklepović Peričić, Joanna Zajac, Anke Rohwer, Jitka Klugarova.

**Methodology:** Malgorzata M. Bala, Tina Poklepović Peričić, Joanna Zajac, Anke Rohwer, Jitka Klugarova, Mike Clarke, Taryn Young.

**Software:** Malgorzata M. Bala.

**Writing – original draft:** Malgorzata M. Bala, Tina Poklepović Peričić, Joanna Zajac, Anke Rohwer, Taryn Young.

**Writing – review & editing:** Malgorzata M. Bala, Tina Poklepović Peričić, Joanna Zajac, Anke Rohwer, Jitka Klugarova, Maritta Välimäki, Tella Lantta, Luca Pingani, Miloslav Klugar, Mike Clarke, Taryn Young.

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
