## [Decision Letter · Decision Letter 0]

8 Apr 2021

PONE-D-21-06471

Informing evidence-based health care (EBHC) teaching and learning: an updated overview of systematic reviews

PLOS ONE

Dear Dr. Bala,

Thank you for submitting your manuscript to PLOS ONE. After careful consideration, we feel that it has merit but does not fully meet PLOS ONE’s publication criteria as it currently stands. Therefore, we invite you to submit a revised version of the manuscript that addresses the points raised during the review process.

We look forward to receiving your revised manuscript.

Kind regards,

Tim Mathes

Academic Editor

PLOS ONE

Journal Requirements:

Reviewers' comments:

Reviewer's Responses to Questions

**Comments to the Author**

1. Is the manuscript technically sound, and do the data support the conclusions?

Reviewer #1: Yes

Reviewer #2: Yes

Reviewer #3: Partly

2. Has the statistical analysis been performed appropriately and rigorously? 

Reviewer #1: N/A

Reviewer #2: Yes

Reviewer #3: N/A

3. Have the authors made all data underlying the findings in their manuscript fully available?

Reviewer #1: Yes

Reviewer #2: Yes

Reviewer #3: Yes

4. Is the manuscript presented in an intelligible fashion and written in standard English?

Reviewer #1: Yes

Reviewer #2: Yes

Reviewer #3: Yes

5. Review Comments to the Author

Reviewer #1: Congratulations on a very well conducted and scientifically sound overview of SR's on this important topic area which adds to the existing body of knowledge. The context in terms of the significance of the topic area and need for the review was well considered although it would have been of benefit to further explicate the purpose of the overview as detailed in the final section e.g. p47 overview of SR's rather than an SR. The manuscript mentions that the Medline search strategy was adapted for the search of Epistemonikas - having reviewed the appendices I cannot determine how it was adapted which would be useful to explicitly describe.

To note I could not review/find Fig 4,5,6. Some minor typos/sentence structure issues within the manuscript as follows:

Typo in abstract – Methods AND Findings

P. 9 – check tense and typos

P.30 typo postgraduatesU

P31 rephrase first paragraph

I have no concerns/additional comments in relation to dual publication, research or publication ethics.

Reviewer #2: Thanks for inviting me to review this manuscript. In this article, authors aimed to update an overview of systematic reviews of studies evaluating the effect of teaching evidence-based practice on learners’ knowledge, skills, attitudes and behaviours. The authors found a total of 22 reviews (6 new reviews) and concluded that knowledge improved, and skills and behaviour were partially improved. This is a well written manuscript. Few comments that the authors might consider helping improve the manuscript:

Comments

- Revise the title to be reflect the study objectives – including some or all of the elements of PICO.

- Interesting to see that you have submitted an application for ethics and got an approval – usually this is not needed.

- Authors might consider justifying their choice to search only Epistemonikos (and deviate from their protocol).

- Authors might consider reassess the RoB for the SRs included in their original review.

- For data analysis, authors might consider this guidance for reporting the results https://www.bmj.com/content/368/bmj.l6890

- I wonder if authors would like to comment about whether it is expected from included reviews to do a quantitative synthesis of their results because of the expected heterogeneity (and therefore assess publication bias) – as described in the assessment of quality of included reviews.

- It is not clear why authors opt to describe the results of the update first and then the overall results. Authors might reconsider this as there is no logical differences between reviews in these two groups.

- Table 3a & 3b are very interesting but authors might need to clearly explain the process behind the decisions regarding the colour coding - have they looked an primary studies within included reviews or just at the conclusion of the review? What about overlapping reviews?

- Figures 2-6 look very interesting but very difficult to read.

Reviewer #3: The issue of EBHC teaching and learning strategies is of major importance and this manuscript has high scientific significance. Overall, the study design is appropriate to answer the aim nut there are some flaws in the text and some points had better be more clearly described. Please explain the search in detail there is no information neither regarding the key words (MESH terms search or not) nor the search algorithm the researchers followed in order to conduct the search in the databases. The whole search process is not clear to me, as many details are not included. Also, valuable already published information on the topic is not included and very important studies like the following are ignored by the authors. Additionally, the introduction is weak and the discussion as well have to be enriched as in the international literature there are many studies investigating this topic. For example: https://pubmed.ncbi.nlm.nih.gov/32878256/

https://pubmed.ncbi.nlm.nih.gov/32905986/

https://pubmed.ncbi.nlm.nih.gov/27649902/

Conclusively, I recommend a major revision of the manuscript.

6. PLOS authors have the option to publish the peer review history of their article (what does this mean?). If published, this will include your full peer review and any attached files.

Reviewer #1: No

Reviewer #2: No

Reviewer #3: No

---

## [Author Response · Author response to Decision Letter 0]

23 May 2021

Comment: 1. Is the manuscript technically sound, and do the data support the conclusions?

Reviewer #1: Yes

Reviewer #2: Yes

Reviewer #3: Partly

Response: Thank you. We appreciate the comments from the reviewers, which we have used to improve the quality of the manuscript as indicated below.

Comment:Reviewer #1: Congratulations on a very well conducted and scientifically sound overview of SR's on this important topic area which adds to the existing body of knowledge. The context in terms of the significance of the topic area and need for the review was well considered although it would have been of benefit to further explicate the purpose of the overview as detailed in the final section e.g. p47 overview of SR's rather than an SR. 

Response: Thank you. We have revised the part of the introduction section regarding justification for this updated review. The paragraph now reads as follows: 

“An update of the overview was needed to follow up on the SRs conducted after the publication of the original overview, which provided clear guidance for future studies about the target interventions, populations, and outcomes, along with preferred ways of measuring them and the required follow-up time, was published. Assessing the subsequent evidence and adding up the available findings will provide a clearer understanding of what works for whom and under which circumstances. We decided to follow the approach of the 2014 overview and to use the already existing evidence from available SRs instead of duplicating the effort and generating needless research waste. We therefore aimed to update the overview published in 2014 to assess the most recent evidence on the effects of various approaches used in teaching EBHC to healthcare professionals at undergraduate and postgraduate level on changes in knowledge, skills, attitudes and behaviour.”

Comment: The manuscript mentions that the Medline search strategy was adapted for the search of Epistemonikas - having reviewed the appendices I cannot determine how it was adapted which would be useful to explicitly describe.

Response: Thank you. We have added information on the search terms used in both versions of the overview, added the MEDLINE search strategy and explained how it was adapted to Epistemonikos. 

We also modified the description in the methods section as follows: “We searched Epistemonikos (Epistemonikos. Epistemonikos Foundation, Arrayán 2735, Providencia, Santiago, Chile; available at https://www.epistemonikos.org) to identify eligible SRs. Epistemonikos is based on searches of a number of relevant databases for systematic reviews (https://www.epistemonikos.org/en/about_us/methods), including: Cochrane Database of Systematic Reviews, PubMed, EMBASE, CINAHL, PsycInfo, LILACS, Database of Abstracts of Reviews of Effects, The Campbell Collaboration online library, the Joanna-Briggs Institute (JBI) database of Systematic reviews and Implementation Reports (JBI Evidence Synthesis), and the EPPI-Centre Evidence Library. We limited our searches for this update to Epistemonikos rather than using the individual databases searched in the previous overview because these are now covered in Epistemonikos, single reliable database. This approach allows reducing the time and resources, as there is no need of duplicates removal, and no influence of possible differences on search terms in individual databases.The MEDLINE search strategy used in the original overview was adapted for the search of Epistemonikos (File S4. MEDLINE and Epistemonikos search strategies). Terms used in the MEDLINE search as index terms and text words were used for searches in Epistemonikos as text words (for example, 1. evidence based healthcare, evidence based medicine, evidence based practice, also with specific medical fields; 2. medical education, teaching, learning, instructions, education) in the titles and abstracts of the records. Instead of keywords for systematic reviews, we used the filters for systematic review available in Epistemonikos.”

Comment: To note I could not review/find Fig 4,5,6.

Response: Thank you. All figures were enclosed in the manuscript submission, it is possible to view the figure as tiff file in more details after downloading original figures using link in the upper right of the page. Figures 4-6 were on pages 57 to 59 of the submission.

Comment: Some minor typos/sentence structure issues within the manuscript as follows:

Typo in abstract – Methods AND Findings

P. 9 – check tense and typos

P.30 typo postgraduatesU

P31 rephrase first paragraph

I have no concerns/additional comments in relation to dual publication, research or publication ethics.

Response: Thank you. We corrected the typos.

Comment: Reviewer #2: Thanks for inviting me to review this manuscript. In this article, authors aimed to update an overview of systematic reviews of studies evaluating the effect of teaching evidence-based practice on learners’ knowledge, skills, attitudes and behaviours. The authors found a total of 22 reviews (6 new reviews) and concluded that knowledge improved, and skills and behaviour were partially improved. This is a well written manuscript. Few comments that the authors might consider helping improve the manuscript:

Comments

- Revise the title to be reflect the study objectives – including some or all of the elements of PICO.

Response: Thank you. We changed the title to: What are the effects of teaching Evidence-Based Health Care (EBHC) at different levels of health professions education? An updated overview of systematic reviews

Comment: - Interesting to see that you have submitted an application for ethics and got an approval – usually this is not needed.

Response: Thank you for your comment. Yes, the protocol for the original overview was approved by Stellenbosch University Research Ethics Committee, as reported in the Methods section, because it was the first chapter of Prof. Taryn Young’s PhD dissertation. 

Comment: - Authors might consider justifying their choice to search only Epistemonikos (and deviate from their protocol).

Response: Thank you. We reported the modifications made in the search in the Differences between the protocol for the primary overview and update of the overview section on page 45-46. We added additional justification to the methods section, so that paragraph reads as follows:

“We searched Epistemonikos (Epistemonikos. Epistemonikos Foundation, Arrayán 2735, Providencia, Santiago, Chile; available at https://www.epistemonikos.org) to identify eligible SRs. Epistemonikos is based on searches of a number of relevant databases for systematic reviews (https://www.epistemonikos.org/en/about_us/methods), including: Cochrane Database of Systematic Reviews, PubMed, EMBASE, CINAHL, PsycInfo, LILACS, Database of Abstracts of Reviews of Effects, The Campbell Collaboration online library, the Joanna-Briggs Institute (JBI) database of Systematic reviews and Implementation Reports (JBI Evidence Synthesis), and the EPPI-Centre Evidence Library. We limited our searches for this update to Epistemonikos rather than using the individual databases searched in the previous overview because these are now covered in Epistemonikos, single reliable database. This approach allows reducing the time and resources, as there is no need of duplicates removal, and no influence of possible differences on search terms in individual databases.The MEDLINE search strategy used in the original overview was adapted for the search of Epistemonikos (File S4. MEDLINE and Epistemonikos search strategies). Terms used in the MEDLINE search as index terms and text words were used for searches in Epistemonikos as text words (for example, 1. evidence based healthcare, evidence based medicine, evidence based practice, also with specific medical fields; 2. medical education, teaching, learning, instructions, education) in the titles and abstracts of the records. Instead of keywords for systematic reviews, we used the filters for systematic review available in Epistemonikos.”

Comment:- Authors might consider reassess the RoB for the SRs included in their original review.

Response: Thank you. We have revised RoB for the studies included in the original overview by using AMSTAR 2 and changed the descriptions in our methods and results sections and figure 2 accordingly.

Comment: - For data analysis, authors might consider this guidance for reporting the results https://www.bmj.com/content/368/bmj.l6890

Response: Thank you. We have prepared a SWiM checklist as supplementary file 3 and added required information in the text of the manuscript. Some of the items are not fully applicable to the results and findings available from systematic reviews included in this overview, which was indicated in the checklist.

Comment: - I wonder if authors would like to comment about whether it is expected from included reviews to do a quantitative synthesis of their results because of the expected heterogeneity (and therefore assess publication bias) – as described in the assessment of quality of included reviews.

Response: Thank you. Due to high heterogeneity of populations included, teaching methods and their comparators, differences in the methodological approach of outcomes assessment and how results are described in narrative way - such analysis in the overview may not be informative, but are not impossible because some of the reviews attempted quantitative analyses. However, many reviews did not have sufficient number of studies per outcome to assess publication bias and did not refer to this. In AMSTAR 1, the instructions suggested using response “Yes” if the review explained that publication bias could not be assessed because there were fewer than 10 included studies, while in AMSTAR 2 there is possibility of response “No meta-analysis”. Since we have now assessed the quality of all reviews using AMSTAR 2 we clarified this issue in the description as follows: “In the SRs included in the original version of the review, lack of protocol was also a common reason for low quality. Among all the included SRs, only one SR (8) conducted a quantitative synthesis and investigated publication bias, while a second SR (36) performed funnel plot analysis despite having conducted no meta-analysis. The justification for not combining studies in meta-analysis in 21 reviews was high heterogeneity in the populations included, teaching methods and their comparators and differences in the methodological approach of outcomes assessment.”

Comment: - It is not clear why authors opt to describe the results of the update first and then the overall results. Authors might reconsider this as there is no logical differences between reviews in these two groups.

Response: Thank you. We moved the text on the results of newly identified reviews from the manuscript to Appendix 8, leaving only the integrated description of the results of all reviews in the main text.

Comment: - Table 3a & 3b are very interesting but authors might need to clearly explain the process behind the decisions regarding the colour coding - have they looked an primary studies within included reviews or just at the conclusion of the review? What about overlapping reviews?

Response: Thank you. We added clarification in the methods section: 

“We used tabular and graphical methods to present the findings at the review level (as reported by the authors of the review) and on the individual study level findings, including information on different population groups, interventions used, and relevant outcomes assessed. 

We used colour coding corresponding to the different directions of effect: dark green (consistent improvement reported by all reviews in a comparison, or all studies if only a single review was included for the comparison), light green (less consistent improvement (improvement found in some reviews/ studies/ certain designs but not in all reviews/ studies/ designs, or improvement found only in a single study with weak design (BA)), yellow (reviews included in the comparison or studies included in the review in the case of a single review reported no difference between the groups), grey (unclear, inconsistent results) and white (not assessed). Individual reviews, not the primary studies included in reviews, were used as the unit of analysis unless only a single review was available for a specific comparison. In coding the colours, we did not take account of any overlap of the primary studies. Explanations of the colour coding are provided alongside tables and figures to allow better understanding of the summarised data.”

Comment: - Figures 2-6 look very interesting but very difficult to read.

Response: Thank you. We have added instruction on how to read as follows: “We used coloured symbols, namely green arrows and yellow circles, to provide a simple display of the direction of the effect, either improvement or no change in each of the specified outcomes. Explanation of the coloured symbols and abbreviations used are available alongside the figures.”.

Comment: Reviewer #3: The issue of EBHC teaching and learning strategies is of major importance and this manuscript has high scientific significance. Overall, the study design is appropriate to answer the aim nut there are some flaws in the text and some points had better be more clearly described. 

Please explain the search in detail there is no information neither regarding the key words (MESH terms search or not) nor the search algorithm the researchers followed in order to conduct the search in the databases. The whole search process is not clear to me, as many details are not included.

Response: Thank you. We clarified the use of text words for searching Epistemonikos. 

The search strategy used to identify SRs in previous version of the overview is presented in Appendix 1 along with the search strategy implemented in Epistemonikos for this update. In this revision of our manuscript , we have added information about the search terms used in the first version of the review and the full MEDLINE search strategy and explained the changes made in adapting the search strategy from MEDLINE to Epistemonikos as follows: “The MEDLINE search strategy used in the original overview was adapted for the search of Epistemonikos (File S4. MEDLINE and Epistemonikos search strategies). Terms used in the MEDLINE search as index terms and text words were used for searches in Epistemonikos as text words (for example, 1. evidence based healthcare, evidence based medicine, evidence based practice, also with specific medical fields; 2. medical education, teaching, learning, instructions, education) in the titles and abstracts of the records. Instead of keywords for systematic reviews, we used the filters for systematic review available in Epistemonikos. There were no restrictions on language of publication. Publication type and Date of publication filters were applied to restrict the search to SRs published from 1 January 2013 to the date of the search (27 October 2020). Ongoing SRs were identified from searches of PROSPERO, Cochrane Database of Systematic Reviews, JBI Evidence Synthesis, Campbell Library and The Best Evidence Medical Education (BEME) Collaboration. Backwards searching was conducted to check for potentially eligible reviews not identified through database searching.”

Comment: Also, valuable already published information on the topic is not included and very important studies like the following are ignored by the authors. Additionally, the introduction is weak and the discussion as well have to be enriched as in the international literature there are many studies investigating this topic. For example: https://pubmed.ncbi.nlm.nih.gov/32878256/

https://pubmed.ncbi.nlm.nih.gov/32905986/

https://pubmed.ncbi.nlm.nih.gov/27649902/

Conclusively, I recommend a major revision of the manuscript.

Response: Thank you. As explained in the manuscript 39 studies were excluded because they did not meet the criteria for a systematic review pre-specified in our protocol. These criteria included having predetermined objectives and predetermined criteria for eligibility (for an update defined as protocol), having searched at least two data sources, of which one was an electronic database and having data extraction and risk of bias assessment performed. All of those 39 reviews did not refer to having a priori defined design, in the update specified as protocol or had not been registered in PROSPERO. None of the three studies suggested by the reviewer meet the inclusion criteria for our overview. We added references to two of the studies in the introduction section (scoping review and the descriptive study) and the third one is referenced in a table of excluded studies.

---

## [Decision Letter · Decision Letter 1]

22 Jun 2021

What are the effects of teaching Evidence-Based Health Care (EBHC) at different levels of health professions education? An updated overview of systematic reviews

PONE-D-21-06471R1

Dear Dr. Bala,

We’re pleased to inform you that your manuscript has been judged scientifically suitable for publication and will be formally accepted for publication once it meets all outstanding technical requirements.

Kind regards,

Tim Mathes

Academic Editor

PLOS ONE

Additional Editor Comments (optional):

Reviewers' comments:

Reviewer's Responses to Questions

**Comments to the Author**

1. If the authors have adequately addressed your comments raised in a previous round of review and you feel that this manuscript is now acceptable for publication, you may indicate that here to bypass the “Comments to the Author” section, enter your conflict of interest statement in the “Confidential to Editor” section, and submit your "Accept" recommendation.

Reviewer #1: All comments have been addressed

2. Is the manuscript technically sound, and do the data support the conclusions?

Reviewer #1: Yes

3. Has the statistical analysis been performed appropriately and rigorously? 

Reviewer #1: Yes

4. Have the authors made all data underlying the findings in their manuscript fully available?

Reviewer #1: Yes

5. Is the manuscript presented in an intelligible fashion and written in standard English?

Reviewer #1: Yes

6. Review Comments to the Author

Reviewer #1: (No Response)

7. PLOS authors have the option to publish the peer review history of their article (what does this mean?). If published, this will include your full peer review and any attached files.

Reviewer #1: No

---

## [Editor Report · Acceptance letter]

6 Jul 2021

PONE-D-21-06471R1 

What are the effects of teaching Evidence-Based Health Care (EBHC) at different levels of health professions education? An updated overview of systematic reviews 

Dear Dr. Bala:

I'm pleased to inform you that your manuscript has been deemed suitable for publication in PLOS ONE. Congratulations! Your manuscript is now with our production department. 

Kind regards, 

on behalf of

Dr. Tim Mathes 

Academic Editor

PLOS ONE